# Self-Determined Motivation Mediates the Association between Self-Reported Availability of Green Spaces for Exercising and Physical Activity: An Explorative Study

Migle Baceviciene [1],* and Rasa Jankauskiene [2]

1  Department of Physical and Social Education, Lithuanian Sports University, Sporto 6,
   44221 Kaunas, Lithuania
2  Institute of Sport Science and Innovations, Lithuanian Sports University, Sporto 6, 44221 Kaunas, Lithuania;
   rasa.jankauskiene@lsu.lt
*  Correspondence: migle75@gmail.com; Tel.: +370-37-302638

**Abstract:** The aim of the study was to test the associations between the self-reported access to exercise in green spaces (GS) and moderate-to-vigorous physical activity (MVPA) testing the mediating role of the motivation. Based on self-determination theory (SDT), we expected that self-determined motivation will mediate the associations between the self-reported availability of GS for exercising (GSE) and MVPA with the most self-determined exercise regulation forms (identified and intrinsic motivation) demonstrating the strongest positive associations between the variables. **Method:** The sample consisted of 2154 participants (74.7% women). The ages ranged from 18 to 79 years, with a mean age of 32.6 (SD = 12.2) years. Participants completed the Behavior Regulation in Exercise Questionnaire-2, the measures of self-reported distance to residential GS (RGS), availability of the GS for exercising (GSE), and physical activity (PA). Logistic regression and path analysis were used to test the associations between study variables. **Results:** Higher reported distance to RGS was associated with lower reported availability of GSE, but not PA. Availability of GSE was directly associated with more frequent MVPA. More autonomous forms of exercise behavior regulation (intrinsic and identified regulations) mediated the associations between self-reported availability of GSE and MVPA. Internal and identified exercise regulations were directly associated with more frequent MVPA. **Conclusions:** The results of the present study support the main tenets of SDT suggesting that self-determined behavioral exercise regulation is an important mediator between the self-reported availability of GSE and general MVPA. Practical implications of these findings are discussed herein.

**Keywords:** physical activity; self-determination; nature orientation; environment; exercise; policy

## 1. Introduction

Recent studies concluded that access to natural green spaces (GS) is associated with the better public health in western societies [1,2]. Exposure to natural GS has been associated with many beneficial health effects such as reduced all-cause and cardiovascular mortality, improved mental health, stress reduction, social cohesion, and quality of life [1,3–10]. Studies have increasingly demonstrated that exposure to GS is associated with connectedness to nature through feelings of non-human form of relatedness, mindfulness and eudaimonia [11–13]. Studies have suggested that access to GS might help to decrease sedentary behavior and to increase physical activity (PA) especially in highly urbanized cities and for economically deprived populations [14,15].

Positive effects of PA on health are well-documented [16]. Nevertheless, sedentary behavior and physical inactivity are major public health concerns [17]. Studies have sug-gested green exercise, or PA while directly exposed in GS, might bring additional benefits to health beyond those of indoor PA [18]. There is some evidence that exercising in nature

helps psychological restoration and helps to prevent unhealthy habits while providing more enjoyment, a sense of life satisfaction, and connectiveness to nature [19–22].

Researchers and urban planners have constantly been trying to understand the forces that drive populations to use GS for recreation [15,23–27]. It has been concluded that availability of GS in living environment is associated with the higher GS contact especially in high urbanization cities [14,25,26,28]. However, studies have suggested that proximity to GS did not affect engagement in PA [29]. In contrast, studies have demonstrated that perceived availability of GS, perception of safety and green qualities were associated with the greater exposure to and exercise in GS [26,30,31].

Environmental and behavioral factors have been frequently analyzed while trying to explain the associations between proximity of GS, perceived availability of GS and PA in GS [32–34]. Studies analyzing the associations between the residential availability of GS and moderate-to-vigorous PA (MVPA) are inconclusive. Some demonstrated that GS availability is associated with the greater MVPA [26,35], however, other studies did not confirm those findings [14,29]. Thus, one of the aims of the present study is to provide more knowledge on this issue.

Further, little is known about how the individual determinants influence exercising in GS [32]. The results from recent studies have suggested that nature orientation, mindfulness, and self-determined motivation might be important driving forces of GS exposure and exercise in GS [11–13,29]. Based on the self-determination theory (SDT) [36], it was previously hypothesized that the connection to nature satisfies the psychological need of relatedness. Nature connection might be linked to intrinsic inspirations, for example, relational emotions as love, care, more pro-environmental decision making and non-human form of relatedness [11]. Unfortunately, motivation has rarely been analyzed in GS studies. Thus, the present study aims to fill this gap.

SDT [36] is an empirically derived theory of human motivation and personality in social contexts that differentiates motivation in terms of being autonomous and/or controlled. SDT is constantly been used to understand motivation for exercise behavior [37–39]. The main idea of the SDT is its consideration of the extent to which the regulation of the behavior has become internalized into the person's sense of self. According to the SDT, varying forms of motivation represent qualitatively different ways in which the behavior is regulated. The theory states that these forms of regulation lie along the continuum ranging from completely non-self-determined to completely self-determined regulation. The three main forms of motivation are described as: amotivation, external motivation and autonomous motivation. Amotivation is the state of lacking any intention to engage to particular behavior. External motivation comprises of external, introjected, identified and integrated regulations. External regulation reflects behavior controlled by external pressures or the behavior with the aim to achieve the externally imposed rewards. In the introjected regulation, the external controlling factors are internalized and are then applied through the self-imposed pressures to avoid guilt and/or to maintain self-esteem. In identified regulation, the outcomes of the behavior become personally valued and the behavior is consciously accepted. Integrated regulation reflects a higher degree of internalization and engaging in the behavior is fully congruent with one's sense of self. Autonomous motivation is fully self-determined motivation with intrinsic regulation. Intrinsic motivation means that a person is taking part in activity for the enjoyment and satisfaction inherent in engaging in the behavior itself. Greater self-determined motivation is linked to better fulfilment of the main psychological needs: autonomy, competence and relatedness [38]. It has been demonstrated that more self-determined or autonomous motivations are associated with better health and well-being outcomes including higher PA, while less self-determined forms of motivation and more external exercise goals are associated with lower well-being outcomes [38,40,41]. Studies have demonstrated that behavioral goals such as improving health, socializing and enjoying the activity are more strongly associated with intrinsic regulation [40], while attractiveness, financial goals and fame are associated with less-self determined behavioral regulation [38,40].

The first aim of the study was to test the associations between the distance to RGS, self-reported availability of GSE and MVPA. The second aim of the present study was to examine the mediating role of motivation in the associations between the availability of GSE and MVPA. To our best knowledge, it is one of the first studies exploring motivation as an individual factor for exposure to RGS and physical activity. In the present study we expected that self-determined motivation would mediate the associations between the availability of GSE and MVPA with the most self-determined exercise regulation forms (identified and intrinsic motivation) demonstrating the strongest positive associations between the variables.

## 2. Methods

### 2.1. Participants

A mixed-gender sample ($n$ = 2154) of Lithuanian adults aged 18–79 participated in this study: 545 study participants were men (25.3%) and 1609 were women (74.7%). The mean age of the sample was $32.6 \pm 12.2$ years. The sample characteristics are presented in Table 1.

**Table 1.** Sample characteristics ($n$ = 2154).

| Characteristics | | $n$ | % |
|---|---|---|---|
| Gender | male | 545 | 25.3 |
| | female | 1609 | 74.7 |
| Residence | urban | 1900 | 88.2 |
| | rural | 254 | 11.8 |
| Education | secondary/apprenticeship | 667 | 31.0 |
| | high | 1487 | 69.0 |
| Body mass index | underweight | 89 | 4.1 |
| | normal weight | 1398 | 64.9 |
| | overweight | 495 | 23.0 |
| | obesity | 172 | 8.0 |
| MVPA (meeting WHO recommendations) | no | 1268 | 58.9 |
| | yes | 886 | 41.1 |
| Distance to RGS | <2 km | 1576 | 73.2 |
| | ≥2 km | 567 | 26.3 |
| Availability for GSE | I have no access | 117 | 5.4 |
| | I have access, but never exercise in GS | 665 | 30.9 |
| | I have access and exercise in GS | 1372 | 63.7 |

WHO = World Health Organization, PA = physical activity, MVPA = moderate-to-vigorous physical activity; RGS = residential green spaces; GSE = green spaces for exercise.

### 2.2. Procedure and Ethics

Data were collected using a cross-sectional online survey distributed through social network and covering major Lithuanian cities and districts. The target group was general population aged 18 years and older; only permanent residents of Lithuania were recruited. The advertisement with the invitation to participate in the survey was shown on Facebook targeting the five largest cities in the country and the 100 km area around them. As part of a more extensive study, respondents completed self-report questionnaires measuring PA, motivation to exercise, and self-reported availability to exercise within GS. A total of 2166 questionnaires were received, from which 12 individuals refused to participate in the survey. Thus, 2154 questionnaires were approved for statistical analysis. The final dataset contained no missing cases.

Prior to the study, permission from Lithuanian Sports University Research Ethics Board was obtained (protocol No. SMTEK-44). No any information letting to identify study participants was collected, thus the anonymity was ensured. Following the Declaration of Helsinki ethical and legal principles of the research, study participants were introduced to the aim of the study. Only participants who agreed to participate in the survey were provided with questionnaires.

*2.3. Measures*

Demographic data. Participants were asked to specify their gender, age, education and place of residence. The sample characteristics are presented in Table 1.

Body mass index (BMI) was calculated in $kg/m^2$ from self-reported height and weight. BMI was classified into four body mass categories: underweight (BMI < 18.5 $kg/m^2$), normal weight (BMI = 18.5–24.9 $kg/m^2$), overweight (BMI = 25.0–29.9 $kg/m^2$) and obese (BMI $\geq$ 30.0 $kg/m^2$) [42]. The majority of the sample (50.1% of men and 69.9% of women) were of normal weight (BMI ranged from 14.7 to 55.6 (M = 23.8, SD = 4.2) $kg/m^2$). The results showed that 40.2% of men and 17.2% of women were overweight, and 9.0% and 7.6%, respectively, were obese. 4.1% of the study population was attributed to the underweight group.

Physical activity (PA) was assessed by Physical Activity Scale (PAS-1) [43]. After obtaining authors' permission, the instrument was translated into Lithuanian language. PAS-1 contains 9 PA levels of different intensities from sleep (0.9 MET) or sedentary behavior to strenuous PA (>6 METs). Each level (A = 0.9 MET, B = 1.0 MET, C = 1.5 METs, D = 2.0 METs, E = 3.0 METs, F = 4.0 METs, G = 5.0 METs, H = 6.0 METs and I $\geq$ 6 METs) was described by providing examples of PA in particular MET level and by small drawings. The questionnaire was constructed asking to indicate duration in hours (1–10) and/or minutes (15, 30 or 45) of each PA level so that the total time comprise 24 h of the typical weekday including sleep.

Frequency of MVPA was assessed to understand if respondents met World Health Organization (WHO) PA recommendations. Study participants were asked to indicate the number of weekdays when they were involved in MVPA at least for 30 min [44]. Those who indicated 4 days or more per week of MVPA were categorized as meeting the WHO PA recommendations.

Distance to RGS was assessed using a single question asking to indicate the average distance from their living place to GS. Possible answers were: "nearby my home"; "0–1000 m from my home"; "2–4000 m from my home"; "5–10,000 m from my home"; "10000m or more from my home"; "I don't know"). Based on previous findings indicating that larger buffer zones up to 2000 m better predict physical health outcomes compared to smaller ones [28], we categorized the sample into two groups: distance to RGS < 2000 m and distance to RGS $\geq$ 2000 m.

Availability of GSE was assessed using a single question "Do you have access to exercise in the park or forest during your leisure time?" with three response options: "I have no access to exercise in GS," "Yes, I have access, but never exercise in GS," "Yes, I have access, and exercise in GS."

The Behavioral Regulation in Exercise Questionnaire 2 (BREQ-2) [45] is a 19-item, self-reported multidimensional instrument based on the SDT. The BREQ-2 is a widely used instrument in the exercise motivation domain. This instrument has been validated in various languages and cultures of the world reporting good psychometric properties [46–50]. The scale is comprised of five subscales measuring five types of exercise regulation: amotivation, external regulation, introjected regulation, identified regulation and intrinsic motivation. Participants were asked questions assessing why they engage in PA and exercise and their responses were measured on a five-point Likert-type scale ranging from 0 ("not true for me") to 4 ("very true for me").

The translation of the BREQ-2 from English into Lithuanian language was performed by professional translators using the back-translation method in the translation agency of

Kaunas city (Lithuania). The original and translated versions were reviewed by translators, and the final Lithuanian version was approved after the pilot study. The Lithuanian version of BREQ-2-LT demonstrated good psychometric properties (presented in Appendix A).

The original structure of the questionnaire was replicated, the questionnaire demonstrated good internal consistency and test-retest reliability. Testing the internal consistency of the BREQ-2-LT subscales, acceptable Cronbach's α values were detected. For this study, Cronbach's α values were reported to be 0.83 for amotivation, 0.75 for external regulation, 0.68 for introjected regulation, 0.74 for identified regulation, and 0.91 for intrinsic regulation subscales. Negative correlations were observed between amotivation, external motivation and autonomous motivation. Subscales representing different levels of autonomous motivation correlated positively. In addition, confirmatory factor analysis (CFA) revealed the original five-factor structure of the BREQ-2 with acceptable model fit parameters (GFI = 0.91; AGFI = 0.88; CFI = 0.91; TLI = 0.90; RMSEA = 0.07). Next, our study confirmed the convergent validity demonstrating that higher levels of PA are associated with more self-determined exercise behavioral regulations. These results coincide with findings of other studies validating the questionnaire in other cultures [45,46,50,51]. For additional information see Appendix A.

*2.4. Statistical Analysis*

To describe study population frequencies and percentages in variables, subgroups were calculated for categorical variables. Cronbach's α coefficients were used for the evaluation of internal consistency. Correlations between the BREQ-2 subscales were assessed by Spearman's correlation coefficient. To test the predictive power of different BREQ-2 subscales on daily PA, multiple linear regression analysis was performed. Finally, a binary multiple logistic regression analysis model was employed to estimate the effect of different factors on self-reported availability to exercise in the area of GS (dichotomized as 0—"no access or not used" or 1—"yes, used"). The statistical analyses were carried out using IBM SPSS Statistics 26 (IBM Corp., Armonk, NY, USA).

Mediation and confirmatory factor analyses were conducted using AMOS version 24 (Analysis of Momentary Structure, SPSS; Armonk, NY, USA: IBM Corp.) The significance of the direct, indirect, and total effects was assessed with chi-square tests, and the significance of the mediational paths was further confirmed through the bootstrap resampling method, with 2000 bootstrap samples and 95% bias-corrected Cis; effects were considered significant ($p < 0.05$) if zero did not appear in the interval between the lower and the upper limits of the CIs. The goodness of fit of the model was assessed using various good fit values: goodness of fit index, GFI ($0.95 <$ GFI $< 1.00$); the adjusted goodness of fit index, AGFI ($0.90 <$ AGFI $< 0.95$); the comparative fit index, CFI ($0.95 <$ CFI $< 1.00$); the Tucker Lewis Index, TLI ($0.95 <$ TLI $< 1.00$); and the root of the mean square error of approximation, RMSEA ($0.00 <$ RMSEA $< 0.05$ as good, $0.05 <$ RMSEA $< 0.08$ as acceptable).

## 3. Results

The sample characteristics are presented in Table 1. Since the data were obtained from online study, we compared the main sample characteristics to the general characteristics that had been reported in other populational studies of Lithuania. As a result of the online information collection and low participation rate among men, women comprised 74.7% of the sample. Other findings suggested that PA and BMI data in our sample reflect the findings of populational studies. For example, according to national survey data, 40.5% of country inhabitants met the WHO PA recommendations which agree with our findings [52]. According to country-representative data, the prevalence of obesity in Lithuanians ages 20–64 is 19.2%. By age group, the obesity rates (men, women) are as follows: ages 20–24 (3.6%, 0.0%), ages 25–34 (8.8%, 6.0%), ages 35–44 (21.8%, 26.9%), ages 45–54 (25.7%, 26.9%), and ages 55–65 (31.6%, 30.3%; Klumbiene et al., 2015). In our survey, the prevalence of obesity by age group (men, women) was the following: ages < 25 (4.3%, 3.9%), ages 25–34 (5.4%, 6.0%), ages 35–44 (7.3%, 8.0%), ages 45–54 (20.3%, 13.2%), and ages ≥ 55 (31.6%,

25.3%). As our study sample consisted of younger study participants, overall prevalence of obesity was lower than in country-representative data, but in the age groups, we did not detect any controversial differences.

Testing the associations between availability of GSE and study variables, logistic regression was developed. The results showed that the reported distance from RGS $\geq 2$ km decreased self-reported availability of GSE by 49% (Table 2). Higher BMI was associated with decreased odds of self-reported availability for GSE. Moreover, a higher level of daily PA and the mostly autonomous forms of motivation (identified and intrinsic regulations) increased the odds of availability for GSE.

**Table 2.** Associations between availability of green space for exercise, demographic criteria, distance to residential green spaces, daily physical activity, and motivation to exercise (*n* = 2154).

| Characteristics | OR | 95% CI | *p* |
|:---:|:---:|:---:|:---:|
| Age | 1.01 | 1.00–1.02 | 0.115 |
| Body mass index | 0.96 | 0.94–0.99 | 0.006 |
| Female gender | 1.05 | 0.81–1.37 | 0.714 |
| Rural place of residence | 0.79 | 0.57–1.09 | 0.142 |
| Level of education | 1.03 | 0.96–1.11 | 0.348 |
| Distance to RGS $\geq$ 2km | 0.51 | 0.40–0.65 | <0.001 |
| PA (MET-hours) | 1.02 | 1.01–1.03 | 0.001 |
| Amotivation | 0.96 | 0.76–1.21 | 0.720 |
| External | 1.11 | 0.96–1.29 | 0.175 |
| Introjected | 0.93 | 0.83–1.05 | 0.233 |
| Identified | 1.34 | 1.08–1.67 | 0.008 |
| Intrinsic | 1.32 | 1.11–1.57 | 0.002 |

OR = odds ratio, CI = confidence interval; PA = physical activity; RGS = residential green spaces; MET = metabolic equivalent.

A series of path analyses were conducted to explore the association between availability for GSE and MVPA. The associations between variables were mediated by the mostly self-determined exercise behavior regulations (intrinsic and identified regulations). Controlled motivation and amotivation were removed from the final model, as they were not significant. The final model predicting frequency of MVPA revealed that all path coefficients were statistically significant at the level *p* < 0.001, and presented an excellent model fit (GFI = 1.00; AGFI = 0.99; TLI = 0.99; CFI = 1.00; NFI = 1.00; RMSEA = 0.031) (Figure 1). The tested model accounted for 33% of the variance of MVPA frequency. Self-reported AGS for exercise directly predicted MVPA frequency. In turn, identified and intrinsic exercise behavioral regulations mediated the effect of self-reported availability of GSE on MVPA frequency. No direct effects between the availability for GSE, autonomous motivation and frequency of MVPA were observed.

Mediated effect of the mostly self-determined exercise regulation forms was also tested, assessing the association between the distance to RGS and self-reported availability for GSE but the model was not significant.

Analysis of indirect effects revealed association between the distance to RGS on frequency of MVPA mediated by self-reported availability for GSE and autonomous motivation (Table 3).

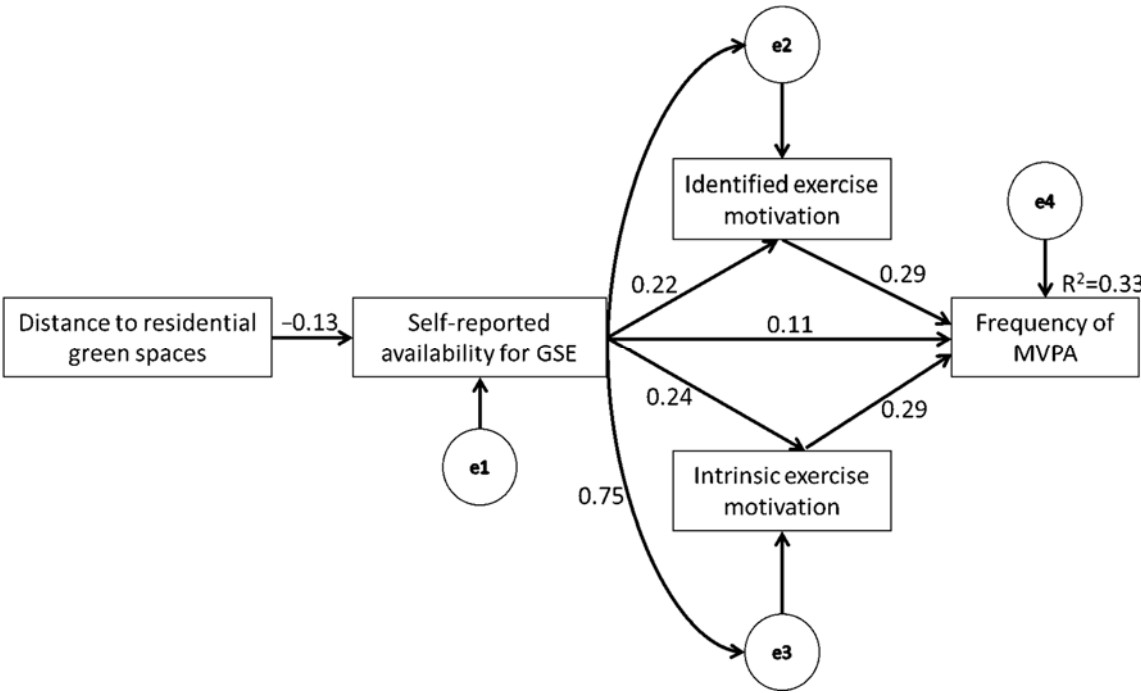

**Figure 1.** Path model on the association between self-reported residential distance to green spaces, availability to exercise in green spaces and physical activity, mediated by autonomous exercise motivation, with standardized estimates. MVPA = moderate to vigorous physical activity; GSE = green spaces exercise.

**Table 3.** Summary of mediation analyses testing the indirect effect between study variables (*n* = 2154).

| Paths | B (95% CI) | *p* | Mediation |
|---|---|---|---|
| Distance to RGS → Availability for GSE → autonomous motivation → frequency of MVPA | −0.049 (−0.067–(−0.063)) | 0.001 | Full |
| Availability of GSE → autonomous motivation → frequency of MVPA | 0.28 (0.24–0.34) | 0.001 | Partial |
| Distance to RGS → Availability for GSE → identified motivation | −0.030 (−0.042–(−0.20)) | 0.001 | Full |
| Distance to RGS → Availability for GSE → intrinsic motivation | −0.037 (−0.051–(−0.025)) | 0.001 | Full |

B = Unstandardized Effect Coefficient; 95% CI = Bootstrapped 95% Confidence Intervals for Unstandardized Effect; *p* = Two-tailed significance.

## 4. Discussion

The aim of the study was to test the associations between the self-reported access to GSE and the frequency of the MVPA testing the mediating role of the motivation. We expected that self-determined motivation would mediate the associations between the self-reported availability of GSE and frequency of MVPA with the most self-determined exercise regulation forms (identified and intrinsic) demonstrating the strongest positive associations between variables. The results of the present study suggested that self-reported availability of GSE was directly associated with MVPA; however, self-determined motivation mediated this association and was responsible for increasing MVPA. The findings of the present study overlap knowledge gained from other studies demonstrating that perceived access to GS is associated with greater MVPA [26]. Reported distance to RGS was not associated with the MVPA in the present study, and these results go in line with similar studies (Ali et al., 2017) suggesting that other factors such as availability, cleanness, safety, aesthetics or greenness might be important for PA in GS [30,31].

However, the important finding of the present study is that individual factors such as exercise motivation is relevant mediator between the self-reported availability of GSE and MVPA. These results suggest that by fostering more self-determined regulation of behavior we can expect greater perception of the availability of GSE and increase of MVPA.

In other words, more internally motivated people might feel GSE is "closer" compared to externally motivated people. Self-determined behavioral regulation is associated with greater mindfulness, life satisfaction and fulfillment of psychological needs (autonomy, competence and relativeness). Thus, non-human nature connectedness might help to fulfill the need for relativeness [11]. The general findings of the present study support the main tenets of SDT and our hypothesis. These results align with the findings of several previous studies suggesting the important role of nature's impact in the GS [13,19,21,34,38,53].

The results of the present study have important practical implications. This study suggests that while self-reported availability of GSE is directly associated with increased MVPA, the self-determined PA motivation makes this association even stronger. It should be mentioned that according to SDT, internal exercise motivation is associated with enjoyment, interest, social interaction and health enhancement goals [40]. Another study also demonstrated that enjoyment was the greater motivator for exercising in GS [54]. Thus, providing high-quality and enjoyable green natural environments, meeting the internal needs of populations with different demographic characteristics, and educational initiatives regarding nature, might increase the public contact to GS and to health enhancing PA.

It should be mentioned that we tried to test the exercise motivation as the mediator between the distance to RGS and self-reported availability of GSE, yet this model did not meet the acceptable statistical values, suggesting that perception of access to GS will not be increased by exercise motivation if the distance from GS is too large. Thus, the intrinsic exercise motivation cannot change the perception of GS availability if access to GS is not available in the living area. Other studies have demonstrated that people rarely travel to GS far from their residence if they do not have available GS in their living areas [14]. This is an important message for urban planners and politicians.

The present study has limitations worth mentioning. First, this study is cross-sectional, which prevents from any causal interpretations between study variables. Further, the measure of availability of GSE is self-reported and respondents who were highly motivated for PA might provide "distorted" opinion about the availability of GSE. In other words, highly motivated respondents might report better availability of GSE and vice versa. However, in this study, respondents reported the distance to RGS (answers were provided in km) and that data adversely correlated with the self-reported availability of the GSE. Therefore, it is reasonable to believe that self-reported availability of GSE was not highly overestimated by exercising people. However, use of the objectively measured data on the availability of GSE is recommended for future studies aiming to understand the role of the individual factors for PA in GS.

Further, intrinsic and identified regulation forms of motivation were intercorrelated, what is another limitation of the study. However, identified behavioral regulation is one of the closest forms of the autonomous or intrinsic motivation, therefore, the correlation between the variables is high. The use of other instruments clearly dividing the autonomous and external motivation might be useful for use in path models of future studies.

Another limitation of the study is that study participants were not selected randomly, and we cannot assert that the sample is country-representative. As respondents were recruited via social network, the participation rate was higher among women than among men. This resulted in the gender imbalance of the sample. Further, nature orientation was not assessed in the present study, thus interpretations about nature orientation and self-determined exercise motivation should be tested in future studies. Finally, we did not perform deeper analysis in terms of demographic, general health satisfaction and other important variables in the present study. These variables might be important understanding green exercise—related attitudes, behavior and motivation. Future studies should address these issues.

## 5. Conclusions

The results of the present study support the main tenets of SDT, suggesting that self-determined behavioral exercise regulation is an important mediator between the self-

reported availability of GSE and MVPA. The present findings suggest that providing better access to high-quality green natural environments and meeting the internal needs of populations with differing demographic characteristics and fostering internal exercise motivation might increase public contact to GS and MVPA, effectively aiding public health. Future studies assessing self-determined exercise motivation and nature orientation are important in understanding the associations between self-reported availability of GSE and MVPA.

**Author Contributions:** Conceptualization, M.B. and R.J.; Data curation, M.B.; Formal analysis, M.B.; Investigation, R.J.; Methodology, M.B. and R.J.; Project administration, M.B.; Software, M.B.; Validation, M.B. and R.J.; Visualization, R.J.; Writing—original draft, R.J.; Writing—review & editing, M.B. and R.J. All authors have read and agreed to the published version of the manuscript.

**Funding:** This research received no external funding.

**Institutional Review Board Statement:** The study was conducted according to the guidelines of the Declaration of Helsinki, and approved by the Institutional Review Board (or Ethics Committee) of Lithuanian Sports University Research Ethics Board (protocol code SMTEK-44, 22 June 2020).

**Informed Consent Statement:** Informed consent was obtained from all subjects involved in the study.

**Data Availability Statement:** The dataset generated and analyzed during the current study is not publicly available. Dataset can be obtained from the corresponding author on reasonable request.

**Conflicts of Interest:** The authors declare no conflict of interest.

## Appendix A

**Table A1.** Internal consistency of the BREQ-2 and correlations between subscales ($n$ = 2154).

| BREQ-2 Motivation Subscales | Cronbach's $\alpha$ | 1 | 2 | 3 | 4 | 5 |
|---|---|---|---|---|---|---|
| Amotivation (1) | 0.83 | 1 | | | | |
| External (2) | 0.75 | 0.33 ** | 1 | | | |
| Introjected (3) | 0.68 | −0.16 ** | 0.19 ** | 1 | | |
| Identified (4) | 0.74 | −0.47 ** | −0.13 ** | 0.48 ** | 1 | |
| Intrinsic (5) | 0.91 | −0.48 ** | −0.25 ** | 0.28 ** | 0.71 ** | 1 |

BREQ-2 = Behavioral Regulation in Exercise Questionnaire 2. ** $p$ < 0.01.

**Table A2.** Associations between daily physical activity and Behavioral Exercise Regulations Questionnaire 2 subscales ($n$ = 2154).

| BREQ-2 Motivation Subscales | B | β | $p$ | VIF |
|---|---|---|---|---|
| Amotivation | 0.43 | 0.03 | 0.392 | 1.59 |
| External | −1.00 | −0.08 | 0.002 | 1.22 |
| Introjected | −0.10 | −0.01 | 0.696 | 1.41 |
| Identified | 1.47 | 0.12 | 0.002 | 2.93 |
| Intrinsic | 2.66 | 0.26 | <0.001 | 2.59 |

BREQ-2 = Behavioral Regulation in Exercise Questionnaire 2; B = unstandardized regression coefficient, β = standardized regression coefficient, VIF = variance inflation factor; model summary: R = 0.36; $R^2$ = 0.13.

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
