# Peer review of "Self-Determined Motivation Mediates the Association between Self-Reported Availability of Green Spaces for Exercising and Physical Activity: An Explorative Study"

_sustainability, doi:10.3390/su13031312_

Round 1

Reviewer 1 Report

The article is, in general, well-written, addresses a relevant topic, clearly outlines the theory, whereby topic and methodology are well-embedded
in literature and the measures well and clearly described. From my perspective, the study deserves to be published.

Major comments:
p3, l133: the description of sampling is insufficient, "through social media" needs further specification. Which or at least what kind of social media?
Was there some link between the channels and the subject? What kind of target groups are the chosen media expected to address? Does this choice explain the
gender imbalance somehow?

My main concern is what looks like a technical detail, but may have important consequences: the distance to GS (green space) is subjectively rated
(as shortly mentioned on p8); this means the distance as well as the perception of what GS is may be distorted (the more I like to exercise, the less
I might bother about the distance). An indication for a similar effect could be the relationship between perceived access and intrinsic motivation in
Table 2, which should be discussed - why are motivated people "closer"? (It may be explainable, e.g. choice of living place or reverse causation, but that's
exactly the consideration I'd like to have added.)
If motivation and subjective perceptions are actually entangled, then the importance of path via motivations could be overestimated,
at the cost of other paths, which is a crucial threat to the interpretations.
Similarly, identified regulation and intrinsic motivation are highly intercorrelated, which should also be discussed as this is not modeled in the path analysis.

Minor comments:
p2,l84: maybe add some more down-to-earth consideration as well, such as aesthetic value of nature
p4,l178: I just want to mention that perceived access to GS is not really a dichotomous question but in fact a continuum (easier - more difficult access).
p5, bottom paragraph: the representativeness consideration is restricted to the variables of interest, obviously. Although global representativeness is not
necessary here, the obvious sociodemographic deviation (gender!) should nevertheless be at least addressed (it later shortly mentioned on p8, but should be
directly stated here as well)

typos and wording:
in general, I recommend one more proofreading round regarding the setting/non-setting of articles
p2, l48: "nevertheless" does not seem to make sense here, no contradiction
l52: "studies have" 2 times
l90: blanc missing before (
l101: sentence structure
l116: AND enjoying
l142: "and thus" reads contradictory

Author Response

Dear Reviewer,

Thank you very much for your time and valuable comments. A detailed point by point table with our answers is attached.

Reviewer 2 Report

L10: the aim of the study (delete general)

L10-12: not sure I understand this, there seems to be grammatical errors

L14: what is a self-determined exercise regulation forms?

L18: ….Questionnaire-2.  In addition, measures of distance between GS and …….

L21: what is a higher proximity?  Does that mean closer to?

L21-22: Going to need to work on grammar here.  It sounds like the closer one was to a GS was associated with a lower perceived access to GS.  That doesn’t make sense

L46: what is a deprived population? Deprived of what?

L50: or PA performed in a GS, (the directly exposed is an awkward way of saying that)

L77: is green exercise the same thing as PA in GS?  If so, you should be consistent with terms

L83: non-human relatedness?  What does this mean especially in context to nature connection. 

L119: shouldn’t use general.  Say something like this: There were three aims to this study.  First, …..

L134: how did it cover major cities? Did you have some way to track this?

Table 1 – I would put in the kg/m2 ranges for each BMI

MVPA – why a range of 4-7?  Why not use ACSM guidelines of at least 5 days a week of MPA (30 min sessions) or 3x/wk of VPA (25 min sessions)

Perceived access – is this the right term?  Because one’s perception may not be consistent with another’s perception.  Say 2 people both live 1000 m away from a GS.  One person may perceive that that is too far?  Also how does perceptions relate to no access, yes not used, or yes used.  Seems like there needs to be a different word than perception

Author Response

(The authors gave the same response as above.)

Round 2

Reviewer 2 Report

The additions/changes made by the authors have improved the manuscript.

The only final comment I have relates to the number of abbreviations.  There are a lot and the use of them may make it challenging for some to read.  The only feedback I would give is to consider using less of them.